# Bivalent EGFR-Targeting DARPin-MMAE Conjugates

**DOI:** 10.3390/ijms23052468

**Published:** 2022-02-23

**Authors:** Lennard Karsten, Nils Janson, Vadim Le Joncour, Sarfaraz Alam, Benjamin Müller, Jayendrakishore Tanjore Ramanathan, Pirjo Laakkonen, Norbert Sewald, Kristian M. Müller

**Affiliations:** 1Cellular and Molecular Biotechnology, Faculty of Technology, Bielefeld University, 33615 Bielefeld, Germany; lennard.karsten@uni-bielefeld.de; 2Organic and Bioorganic Chemistry, Faculty of Chemistry, Bielefeld University, 33615 Bielefeld, Germany; n.janson@uni-bielefeld.de; 3Translational Cancer Medicine Research Program, Faculty of Medicine, University of Helsinki, 00290 Helsinki, Finland; vadim.lejoncour@helsinki.fi (V.L.J.); jk.tanjoreramanathan@helsinki.fi (J.T.R.); pirjo.laakkonen@helsinki.fi (P.L.); 4Biochemistry III, Faculty of Chemistry, Bielefeld University, 33615 Bielefeld, Germany; sarfaraz.alam@uni-bielefeld.de; 5Biofidus AG, 33615 Bielefeld, Germany; benjamin.mueller@biofidus.de; 6Laboratory Animal Center, Helsinki Institute of Life Science (HiLIFE), University of Helsinki, 00290 Helsinki, Finland

**Keywords:** EGFR, DARPin, antibody-drug conjugates, MMAE conjugates, Knoevenagel ligation, formylglycine-generating enzyme, cytotoxicity, in vivo imaging, xenograft, SCC model

## Abstract

Epidermal growth factor receptor (EGFR) is a validated tumor marker overexpressed in various cancers such as squamous cell carcinoma (SSC) of the head and neck and gliomas. We constructed protein-drug conjugates based on the anti-EGFR Designed Ankyrin Repeat Protein (DARPin) E01, and compared the bivalent DARPin dimer (**DD1**) and a DARPin-Fc (**DFc**) to the monomeric DARPin (**DM**) and the antibody derived scFv425-Fc (**scFvFc**) in cell culture and a mouse model. The modular conjugation system, which was successfully applied for the preparation of protein-drug and -dye conjugates, uses bio-orthogonal protein-aldehyde generation by the formylglycine-generating enzyme (FGE). The generated carbonyl moiety is addressed by a bifunctional linker with a pyrazolone for a *tandem* Knoevenagel reaction and an azide for strain-promoted azide-alkyne cycloaddition (SPAAC). The latter reaction with a PEGylated linker containing a dibenzocyclooctyne (DBCO) for SPAAC and monomethyl auristatin E (MMAE) as the toxin provided the stable conjugates **DD1**-MMAE (drug-antibody ratio, DAR = 2.0) and **DFc**-MMAE (DAR = 4.0) with sub-nanomolar cytotoxicity against the human squamous carcinoma derived A431 cells. In vivo imaging of Alexa Fluor 647-dye conjugates in A431-xenografted mice bearing subcutaneous tumors as the SCC model revealed unspecific binding of bivalent DARPins to the ubiquitously expressed EGFR. Tumor-targeting was verified 6 h post-injection solely for **DD1** and **scFvFc**. The total of four administrations of 6.5 mg/kg **DD1**-MMAE or **DFc**-MMAE twice weekly did not cause any sequela in mice. MMAE conjugates showed no significant anti-tumor efficacy in vivo, but a trend towards increased necrotic areas (*p* = 0.2213) was observed for the **DD1**-MMAE (*n* = 5).

## 1. Introduction

The epidermal growth factor receptor (EGFR, ErbB1, HER1) is a member of the ErbB type I receptor tyrosine kinase (RTK) family. Binding of a soluble ligand to the ectodomain of the EGFR induces homo- or heterodimerization, which triggers activation and internalization [1,2,3]. Lysosomal targeting or receptor recycling are the two main destinations upon EGFR internalization [4]. The Human Protein Atlas project reveals ubiquitous expression of EGFR in epithelial, mesenchymal and neuronal cells forming healthy tissue [5]. The EGFR is the most studied RTK due to its general role in signal transduction related to cell proliferation and migration as well as its association with oncogenesis [4,6]. EGFR is an overexpressed and validated target in various cancers such as colorectal, gynecological and urological cancers, squamous cell carcinomas (SSC) of the head and neck, non-small cell lung cancer, renal and breast neoplasms and gliomas [7].

A large variety of EGFR-targeting antibodies, antibody mimetics and peptides has been developed to be used as homing devices with cytotoxic agents in order to reduce systemic side effects of anti-tumor drugs [1]. Designed Ankyrin Repeat Proteins (DARPins) are non-immunoglobulin scaffolds featuring a groove-like target-binding surface with efficient folding properties and high solubility [8]. DARPin E01 (K_D_ = 0.5 nM) was identified by phage display and shown to strongly bind the EGFR ectodomain III with an overlapping epitope of the natural ligand EGF [9,10]. Furthermore, the ability to inhibit EGFR phosphorylation was shown for DARPin E01. On the other hand, the single-chain variable fragment 425 (scFv425, V_H_-(G_4_S)_3_-V_L_), which was derived from the monoclonal murine IgG2a antibody 425, binds a non-overlapping epitope at the same domain of the human EGFR with moderate affinity (200 nM < Kd < 400 nM) but not the murine EGFR [11,12,13,14]. So far, there are four FDA-approved anti-EGFR monoclonal antibodies (cetuximab, panitumumab, nimotuzumab, necitumumab), while one antibody-drug conjugate (ADC) (depatuxizumab mafodotin) against glioblastoma was discontinued in phase III clinical trials in 2019 [15,16].

The first- and second-generation ADCs like depatuxizumab mafodotin were generated by the coupling of cytotoxic drugs to the lysine amino group or the cysteine thiol, which usually results in a heterogeneous mixture of ADCs [17]. As a result, current research is focused on the development and optimization of site-directed conjugation methods leading to a homogeneous ADC formulation with improved pharmacokinetic properties, increasing the therapeutic window [18]. Bio-orthogonal reactions, e.g., using enzymes that catalyze site-directed modifications of short amino acid sequences, should be applied. Besides sortase [19,20], transglutaminase [21,22,23,24,25,26] and farnesyl transferase [27], the formylglycine-generating enzyme (FGE) represents an interesting tool for site-directed modification. FGE oxidizes cysteine within the consensus sequence CxPxR to the non-canonical, electrophilic amino acid C^α^-formylglycine (FGly) [28,29,30,31,32,33,34], which can be addressed selectively by nucleophiles. In particular, hydrazine-functionalized indoles (Hydrazino-*iso*-Pictet-Spengler ligation) [35,36,37,38,39] or pyrazolones (*trapped* and *tandem* Knoevenagel ligation) [40,41,42,43] are used in combination with FGE to label proteins or to produce antibody-drug conjugates.

## 2. Results and Discussion

Existing HIPS and Knoevenagel ligations have been further optimized by utilizing bifunctional HIPS and *tandem* Knoevenagel reagents in combination with highly efficient strain-promoted azide-alkyne cycloadditions (SPAAC) [39,40]. The new coupling strategy was tested with mono- and bivalent DARPins and a scFv construct targeting the EGFR. We focused on three aspects of antibody mimetic drug conjugates. First, the binding moiety and valency toward EGFR was addressed by choosing a previously described scFv and a DARPin, which were fused to an IgG1 Fc antibody domain. Second, improved coupling efficiency was achieved by converting the genetically encoded FGly-tag CxPxR with formylglycine-generating enzyme (FGE) in combination with a novel coupling strategy. The relatively slow reaction of the formylglycine residue in a Knoevenagel reaction generating an azide-modified protein was combined with the fast strain-promoted azide-alkyne cycloaddition (SPAAC) [40] of the DBCO-modified drug monomethyl auristatin E (MMAE). Third, uptake and anti-tumor activity was analyzed in cellular assays and in a mouse xenograft model.

### 2.1. Chemical Synthesis and Protein Conjugation

#### 2.1.1. Synthesis of PEGylated DBCO-MMAE Linker

The twofold conjugation of the hydrophobic MMAE obtained by *tandem* Knoevenagel ligation to the C-terminus of proteins results in increased local hydrophobicity of the protein-drug conjugate because of the neighboring MMAE moieties [42]. This can lead solubility issues and reduced plasma half-life. To overcome this problem, a novel PEGylated DBCO-MMAE linker construct (**6**) was synthesized comprising the dibenzo-azacyclooctyne DBCO for SPAAC, the enzymatically cleavable dipeptide linker valine-citrulline, the self-immolating 4-aminobenzyl group (PAB), and the toxin MMAE. As shown previously, PEGylation (PEG_8-12_) of drug linkers [44] or DARPins [45] significantly improves hydrophilicity and pharmacokinetics while reducing clearance of the conjugate and thus enhancing anti-tumor efficacy.

The synthesis of DBCO-PEG_2_-Lys(mPEG_10_)-βAla-Val-Cit-PAB-MMAE (**6**) was performed by a fragment condensation approach between DBCO-PEG_2_-Lys(mPEG_10_)-βAla-Val-OH (**2**) and H-Cit-PAB-MMAE (**5**) (Figure 1). For this purpose, the PEGylated peptide **1** was synthesized by solid-phase peptide synthesis (SPPS). In this procedure, the PEG linker was coupled on resin to the lysine side chain using HATU/HOAt after acidic cleavage of the Mtt protecting group. The subsequent coupling of the adipic acid-modified DBCO had to be done in solution with the NHS ester. For the synthesis of **5**, the amine of citrulline was Boc-protected. After coupling of 4-aminobenzyl alcohol (PAB-OH) with EEDQ, the alcohol was derivatized with 4-nitrophenyl carbonate, followed by coupling of MMAE in the presence of HOAt. The cleavage of the Boc protecting group was performed with dry TFA due to the observed acid-sensitivity of the carbamate.

The final fragment condensation of **2** and **5** was accomplished with PyAOP/HOAt, since coupling reactions with HATU/HOAt or DEPBT resulted in significantly higher epimerization also in a test reaction between Fmoc-βAla-Val-OH and H-Cit-OMe (see Supporting information chapter 8). Attempts to purify **6** by C18 RP-HPLC in the presence of acid gave insufficient purities of <85% (decomposition of DBCO), while the addition of 0.1% NH_3_ provided purities of >95%.

#### 2.1.2. FGly Formation in Proteins

Different, C-terminally FGly-motiv tagged DARPin E01 variants were obtained as EGFR-targeting proteins: a monomer (**DM**), a dimer with (GGGGS)_4_-linker between both DARPins (**DD1**), a dimer with an additional N-terminal CTPSR (**DD2**), and a DARPinFc fusion protein (**DFc**). In addition, the single-chain fragment of the antibody 425 (scFv425) fused with a human IgG1 Fc fragment (**scFvFc**) was used (see methods and materials for protein design). Enzymatic conversions of the CTPSR sequence were performed using either recombinant bacterial FGE from *Mycobacterium tuberculosis* (MtFGE) or recombinant human FGE (hFGE) (Figure 1A). In contrast to the fully copper-loaded hFGE obtained after purification in insect cells, only 1% of MtFGE is loaded with copper after expression in *E*. *coli* [46]. Therefore, stoichiometric amounts of CuSO_4_ were added during the reaction with MtFGE, resulting in increased activities due to in situ reconstitution [35,40]. Successful conversions were verified for all DARPin constructs by tryptic digestion and MALDI-ToF-MS using CLCCA or DNPH as matrix (Appendix A). For the **scFvFc**, verification by MALDI-ToF MS was not possible, because the desired peptide fragment was not detected.

#### 2.1.3. Protein-MMAE Conjugates

*Tandem* Knoevenagel ligation followed by copper-free SPAAC was used for the chemical conjugation of proteins. Therefore, the bifunctional linker, *tandem* Knoevenagel-azide (**7**), was ligated to the FGly-containing proteins at 30 °C for 20–24 h, resulting in the incorporation of two azido groups per FGly residue (Figure 1A).

Initial SPAAC experiments were performed after rebuffering into PBS with commercial DBCO-PEG_3_-Val-Cit-PAB-MMAE. In the case of **DM**, complete conversion was already achieved with one equivalent as demonstrated by SDS-PAGE (Appendix A). Complete labeling was not achieved with **DD2**, regardless of concentration, stoichiometry, temperature, incubation time, and type of FGE (Appendix A). Therefore, only the DARPin dimer variant with one C-terminal tag (**DD1**) was used in further studies. Precipitation occurred during the conjugation of **DFc** and **scFvFc**, which may be due to the high hydrophobicity of the MMAE conjugates. Conjugation experiments with the PEGylated MMAE linker **6** (Appendix A) resulted in conjugation efficiencies similar to that of the commercial DBCO linker (Appendix A) and provided stable conjugates even at high concentrations of **DM**, **DD1**, and **DFc**. Precipitation, albeit slightly reduced, was still observed for **scFvFc** despite PEGylation, probably due to stability issues of the protein itself. As a result, this **scFvFc** construct appeared to be inappropriate for conjugation on a larger scale.

MMAE conjugates of **DD1** and **DFc** were prepared in preparative amounts. The purification was performed by hydrophobic interaction chromatography (HIC) with a HiTrap Phenyl HP column (Figure 2A), followed by dialysis into PBS (**DD1**: 1.97 mg/mL, **DFc**: 2.25 mg/mL). The masses of the desired conjugates were verified by LC-ESI-ToF MS (Figure 2A). For the **DD1**-MMAE conjugate, one signal (*M*_w,obs_: 41,440.3 Da) was detected corresponding to two bound MMAEs (*M*_w,calc_: 41,438.6 Da). Further signals for FGE, unconjugated material or conjugates with lower drug-antibody ratio (DAR) could not be observed, resulting in a calculated DAR of 2 which represents the maximum of two possible payloads. To analyze the **scFvFc**-MMAE conjugate, the construct was digested with PNGase. The signals observed corresponded to the deglycosylated **scFvFc** (*M*_w,obs_: 100,459.8 Da) and the residual glycosylated **scFvFc** (*M*_w,obs_: 101,903.5 Da) with 4 MMAEs attached (*M*_w,calc_ of deglycosylated protein: 100,449.7 Da). The mass difference of 10.1 Da can be explained by the accuracy of the mass spectrometer. Residual FGE or conjugates with lower DAR could not be detected, resulting in a DAR of 4 with a maximum of four possible payloads. Finally, overall yields for three steps (FGE conversion, *tandem* Knoevenagel ligation, and SPAAC) of 44% and 42% were obtained for the **DD1-** (DAR 2) and **DFc**-MMAE (DAR 4) conjugates, respectively. The superior results underscore that the established modular conjugation system consisting of FGE and *tandem* Knoevenagel ligation in combination with SPAAC is an excellent alternative to previously published methods [24,25,38,41,42], as low amounts of expensive drug-containing reagents are yielding in highly homogeneous drug-conjugates. This bioconjugation strategy may be used for novel homogenous ADCs with higher DAR in the future.

#### 2.1.4. Protein-Alexa Fluor 647 Conjugates

Dye conjugates of **DM**, **DD1**, **DFc** and **scFvFc** were prepared using the commercially available near-infrared dye DBCO-Alexa Fluor 647 to investigate EGFR targeting regarding receptor binding and internalization. All conjugates were purified and fully separated from FGE and excess DBCO dye by HIC (Figure 2B top panels), and then dialyzed against PBS. Analysis of the conjugates by SDS-PAGE (Figure 2B middle, bottom panels) showed that all proteins were successfully derivatized at least once with the fluorophore. Dye labeling was verified by the color of the conjugate in the absence of Coomassie (Figure 2B middle panels) and a shift compared to the unconjugated protein (Figure 2B bottom panels). Protein concentrations were determined by approximation via the absorbance of the chromophore at 650 nm and its extinction coefficient (270,000 M^−1^∙cm^−1^) (Appendix A).

### 2.2. In Vitro Studies

#### 2.2.1. EGFR-Mediated Endocytosis of Alexa Fluor 647 Conjugates

Live cell imaging experiments were conducted with confocal fluorescence microscopy to investigate the impact of the chemical modifications on the protein vehicles **DM**, **DD1**, **DFc** and **scFvFc**. The A431 squamous cell carcinoma or the MDA-MB-231 and MCF7 breast cancer cells with about 2.0 × 10^6^, 2.6 × 10^5^ and 2.0 × 10^3^ EGFR sites per cell [48,49], respectively, were incubated with the dye conjugates for 10 min at 37 °C, or for 4 h at 4 °C. For each dye conjugate, strong binding and internalization signals were observed for A431 but not for MCF7 cells at 37 °C (Figure 3). Increased internalization was observed from 2 to 6 h upon dye-conjugate treatment of A431 cells (Appendix A), indicating that the endocytic pathway was comprised of more late rather than recycling endosomes [50]. MDA-MB-231 cells with intermediate to high EGFR expression also showed binding and internalization of all dye conjugates (Appendix A).

Overlay fluorescence images revealed colocalization with lysosomes upon internalization into A431 cells for all dye conjugates at 37 °C (Appendix A). Internalization was inhibited at 4 °C, suggesting an EGFR-mediated endocytosis [51]. The patient-derived glioblastoma cell line BT12, also featuring EGFR upregulation, was exploited to investigate the penetration depth of dye conjugates into tumor cell spheroids [52]. The penetration depth determined for the fluorescently labeled **scFvFc** was similar to **DD1** and higher than that of the **DFc** (Figure 4). This indicates that penetration depth depends on kinetic parameters like the dissociation rate constant rather than the molecular weight of the protein. These live cell imaging experiments demonstrated that **DM**, **DD1**, **DFc** and **scFvFc** retained the EGFR-binding and receptor-mediated endocytosis after Alexa Fluor 647 conjugation.

#### 2.2.2. MMAE Conjugates with Sub-Nanomolar Cytotoxicity In Vitro

The cytotoxicities of anti-EGFR MMAE conjugates were evaluated against A431 cells derived from squamous cell carcinoma, while the MCF7 breast adenocarcinoma cells and HDFa primary dermal fibroblasts with low EGFR expression levels served as negative controls [53]. Cells were incubated with increasing concentrations of **DD1**-MMAE, **DFc**-MMAE or free MMAE for 72 h at 37 °C and analyzed for remaining metabolic activity using the alamarBlue assay. Similar IC_50_ values between 1–2 nM were observed for all tested cell lines treated with the free MMAE (Figure 3B–D). In comparison, IC_50_ values of 0.80 nM and 0.30 nM were determined for A431 cells incubated with **DD1**-MMAE or **DFc**-MMAE, respectively. In contrast, the anti-EGFR MMAE conjugates exhibited a 100-fold lower cytotoxicity against MCF7 and HDFa cells. A similar specificity had been reported for an anti-EGFR ADC with Val-Cit-PAB-MMAE linker [54].

A few non-proliferating cells, which tolerate a high drug dose in vitro were identified in every experiment using bright-field microscopy (Appendix A). In case of the A431 cells, these remaining cells might express low EGFR levels as the heterogeneity of receptor expression among individual A431 cells has been described before [55]. Increased cytotoxicity of **DFc**-MMAE over **DD1**-MMAE could be explained by its higher DAR. Different orientations and distances between the antigen binding sites of **DFc** and **DD1** might also affect their binding affinities and IC_50_ values. Taken together, in vitro cytotoxicity studies and imaging showed selective EGFR-mediated uptake and drug release for **DD1**-MMAE and **DFc**-MMAE demonstrating functionality of the novel PEGylated DBCO-MMAE linker (**6**).

### 2.3. In Vivo Studies

Correlations between in vitro and in vivo studies for EGFR-targeting DARPin constructs were examined to advance the development of drug-conjugates for preclinical studies. Thus, Rj:NMRI-Foxn1 nu/nu female mice bearing subcutaneous A431 squamous cell carcinomas (SCC) expressing high EGFR levels were used to simultaneously investigate Alexa Fluor 647 dye and MMAE conjugates.

#### 2.3.1. In Vivo Imaging of Anti-EGFR Dye Conjugates

Tumor targeting in vivo was tested for the **DM**-, **DD1**- and **DFc**-dye conjugates (Appendix A). The **scFvFc**-dye conjugate binding the human but not the murine EGFR was used as a positive control [56]. Xenografted mice with a tumor volume between 400–500 mm^3^ were intravenously injected into the tail vein with one single dose of the dye conjugate (10–15 µM) and the biodistribution was followed over time via whole-body imaging (*n* = 2 mice per group). **scFvFc** showed the best tumor-targeting properties among all dye conjugates 6 h post-injection (Figure 5).

The highest fluorescence signal of **scFvFc**-Alexa Fluor 647 on tumor site was measured 24–48 h post-injection (Appendix A). Plenty of unspecific binding was observed for the DARPin-dye conjugates. **DFc** showed excessive unspecific binding possibly due to high EGFR-affinity, which correlates with its in vitro cytotoxicity data. Since EGFR is a ubiquitously expressed receptor, **DFc** is probably binding cells with lower EGFR levels before reaching the tumor site.

Dorsal and lateral views adumbrate blood vessels underneath the skin indicating unspecific binding of **DFc** to endothelial cells (Appendix A). Ventral views show mice with fluorescence accumulation into the bladders 20 min post-injection with **DM**- and **scFvFc**-Alexa Fluor 647 giving evidence for a fast renal clearance (Appendix A). Monomeric DARPins are known for long-time serum stability, but also for their fast blood clearance in mice within minutes due to their low molecular weight [45]. **scFvFc** faced aggregation problems during bioconjugation and storage, which might explain decreased stability in vivo and rapid clearance for a major portion of this conjugate. In contrast, bladder fluorescence 48 h post-injection indicates a later clearance for **DD1** (Appendix A). In vivo imaging revealed unspecific binding for DARPin-dye conjugates compared to the positive control **scFvFc**. A pretherapy of unconjugated EGFR binder could be exploited to saturate the murine EGFR-binding sites and test a possible reduction of unspecific binding upon DARPin-dye conjugate treatment.

#### 2.3.2. Ex Vivo Imaging of Anti-EGFR Dye Conjugates

Ex vivo imaging was performed to quantify the relative enrichment of dye conjugates inside major organs. A431-xenografted mice were PFA-perfused 6 h after **DM**-, **DD1**-, **DFc**- or **scFvFc**-Alexa Fluor 647 injection to preserve dissected organs during imaging. **DM** and **DD1** both showed high and moderate accumulation inside the kidneys and liver, respectively (Figure 6).

Renal accumulation of DARPin-dye conjugates can be attributed to the molecular size in accordance with the literature [57]. The highest fluorescence intensity in the liver and low signal in the kidneys was observed for **DFc**. Previous studies using the PEGylated radionuclide-labeled anti-human EGFR antibody cetuximab (C225) revealed non-specific liver uptake, even though cetuximab does not bind to murine EGFR [58]. Also, hydrophobic moieties of antibody-drug conjugates are known to contribute to unspecific uptake by hepatic non-parenchymal cells like sinusoidal endothelial cells or Kupffer cells [59]. It is known that the mouse liver has elevated EGFR expression [60,61]. This could explain the high accumulation of **DM**, **DD1** and **DFc**, but not **scFvFc** dye conjugates in the liver. In the case of **DFc**, a stronger liver accumulation might be due to higher avidity or hepatic clearance might be enhanced by the fraction of Fc molecules with agalactosylated glycans that interact with cell surface lectins like the mannose receptor [62].

The fluorescently labeled positive control **scFvFc** showed primarily accumulation in the kidneys. All other tissues including tumors showed normalized signals below 10%. However, Alexa Fluor 647 fluorescence from **DD1** and **scFvFc**, but not from **DM** or **DFc**, was detected around blood vessels after podocalyxin staining of fixed cryo sections of tumors (Figure 6F,G and Appendix A). All dye conjugates were detected in cryo sections of kidneys and liver (Appendix A). **scFvFc** and **DM** showed prominent signals at the brush border of proximal tubules. Elevated fluorescence in the kidneys 96 h post-injection with **DFc**-dye conjugate suggested the slowest clearance (Appendix A). Ex vivo and in vivo imaging showed consistent results, but microscopy of cryo sections was required to identify **DD1** and **scFvFc** as the tumor penetrating dye conjugates.

#### 2.3.3. In Vivo Efficacy of MMAE Conjugates

Mice bearing the A431 xenografts were randomized into three treatment groups (*n* = 5 mice per group) with mean tumor volumes of approximately 90 mm. Intravenous injections of 6.5 mg/kg **DD1**-MMAE, **DFc**-MMAE or PBS solution into the tail vein were performed twice a week for 2 weeks. Tumors of the PBS treated control group grew to an average volume of 504 ± 197 mm^3^ at day 42, which was not significantly different from the **DD1**-MMAE (300 ± 52 mm^3^, *p* = 0.0799) or **DFc**-MMAE (374 ± 159 mm^3^, *p* = 0.3328) treated groups (Figure 7A). Drowsiness for 1–3 h upon **DFc**-MMAE injection has been constantly observed as an immediate side effect, suggesting a potential crossing the blood-brain-barrier of the central nervous system. On the other hand, neither loss of body weight nor sudden death occurred, nor apparent abnormalities were found during dissection of mice (Figure 7B,C). Even organ weights among the treatment groups declared safe doses of applied MMAE conjugates (Appendix A).

Cryo sections revealed necrotic areas inside both the treated and control tumors, which is a common feature in solid tumors and caused by ischemic injury [63]. These necrotic areas put an uncertainty on the tumor volume for the tumor growth data (Figure 7D). Cell debris, erythrocytes and larger immune cells were identified inside the pus-like liquid from the necrotic areas using bright-field microscopy (data not shown). A mean necrotic area of 26 ± 11% was determined for tumors in the PBS treated control group, which was not significantly different from the **DD1**-MMAE (43 ± 22%, *p* = 0.2213) or **DFc**-MMAE (29 ± 19%, *p* = 0.8341) treated groups (Appendix A). There is a bias due to the selection of analyzed cryo sections, but for several A431 xenograft models this degree of tumor necrosis is not occurring or not reported [64,65,66]. Tumor interstitial fluid (TIF) is a result of nonfunctional lymph vessels inside the tumor center [67,68]. The correlation of elevated TIF pressure and subcutaneous A431 tumor volume as well as sagging fluid protuberance development has been reported [69,70]. This impedes the volume determination of the actual tumor tissue by using the vernier caliper.

Immunofluorescence imaging of tumor sections revealed continuous EGFR expression and presence of blood vessels throughout the tumor tissue (Figure 7D). The terminal deoxynucleotidyl transferase (TdT) dUTP nick-end labeling (TUNEL) procedure (Appendix A) clearly showed lethal levels of cellular DNA damage, but not increased cell death around blood vessels among the treatment groups. In addition, Ki-67 staining (Appendix A) revealed an intensely proliferative live tumor tissue after MMAE conjugate treatment compared to other therapies like the orally administered vandetanib [65]. Finally, the in vivo studies demonstrated that insufficient amounts of **DD1**-MMAE and **DFc**-MMAE reached the subcutaneous tumor to produce an anti-tumor effect. Nevertheless, the standard caliper method may hamper the detection of smaller changes in the growth of the actual live tumor tissue because of the necrotic areas inside the solid tumors of A431 xenografts.

The commonly used microtubule-targeting agent MMAE, whose pharmacokinetics is well studied, showed moderate anti-tumor efficacy in A431-xenograft models when administered as free drug [54,71]. Other well-studied DARPin-drug conjugates target the tumor-associated antigen epithelial cell adhesion molecule (EpCAM). DARPin Ec4 was genetically fused to exotoxin A (ETA) and non-specific binding was not observed in preclinical studies [72]. Another variant, Ec1, was conjugated to MMAF and half-life extension was achieved by either PASylation or coupling to serum albumin [73,74]. In vivo studies on tumor targeting of anti-EpCAM DARPin showed promising results. This better outcome might be explained by a lower cross reactivity of the respective DARPin to the murine EpCAM and a lower expression of EpCAM in the vasculature and accessible tissues compared to that of EGFR.

## 3. Materials and Methods

### 3.1. Organic Synthesis

Hydrolysis and oxygen-sensitive reactions were performed under argon using the Schlenk technique. Chemicals and solvents were obtained from commercial sources and used without further purification. Dry solvents were distilled and stored over desiccants. Flash column chromatography was performed using 0.040–0.063 mm diameter silica gel (Merck). Thin layer chromatography (TLC) was performed using silica gel 60 F254 on aluminum foil (Merck). Detection was carried out under UV light and/or by staining solutions (bromocresol, potassium permanganate or ninhydrin). HPLC analysis was conducted on the Agilent 1200 series LC-MS system with a Luna C18 column (100 × 2 mm, particle size 3 μm, Phenomenex) using a 5–95% linear acetonitrile gradient in 12 min (+0.1% formic acid). Purification by HPLC was achieved with a LaChrom HPLC system (Merck Hitachi) equipped with a D-7000 interface, L-7520 UV detector, L-7150 pump and a C18 Jupiter column (250 × 21.2 mm, particle size 10 μm, flowrate 10 mL/min, Phenomenex). Analytical and preparative HPLC were monitored at 220 nm. NMR spectra (^1^H, ^13^C and 2D) were measured with a Bruker DRX500 or BRUKER Avance 500 at 298 K and processed with MestreNova (MestreLab, Santiago de Compostela, Spain). Chemical shifts are given relative to the residual solvent peaks and multiplicities are given as s (singlet), d (doublet), t (triplet), q (quartet), and m (multiplet), with coupling constants in Hertz.

**Boc-Cit-OH:** H-Cit-OH (2 g, 11.4 mmol, 1 eq.) and Boc_2_O (2.5 g, 11.4 mmol, 1 eq.) were dissolved in H_2_O (15 mL) and THF (7.5 mL), followed by dropwise addition of a solution of Na_2_CO_3_ (3 g, 28.3 mmol, 2.5 eq.) in THF (5 mL). The reaction mixture was stirred at rt overnight and subsequently warmed to 50 °C. Next, the solution was washed with petroleum ether (2×) and THF was removed under reduced pressure. The pH of the aqueous phase was adjusted with sat. KHSO_4_ to pH 2, extracted with EtOAc (5×) and washed with brine. After drying over Na_2_SO_4_ and removal of the solvent, Boc-Cit-OH (2.72 g, 9.54 mmol, 84%) was obtained as a colorless solid and was used without further purification. ^1^H-NMR (500 MHz, DMSO-*d_6_*) δ = 1.38 (s, 9H, Boc-H), 1.45–1.55 (m, 2H, -CH_2_-,) 1.59–1.69 (m, 2H, -CH_2_-), 2.92 (q, ^3^*J* = 6.42 Hz, 2H, -CH_2_-NH-), 3.84 (m, 1H, Boc-NH-CH-), 5.35 (s, 2H, -CO-NH_2_), 5.91 (t, ^3^*J* = 5.74 Hz, 1H, -CH_2_-NH-), 7.06 (d, ^3^*J* = 8.06 Hz, 1H, Boc-NH-), 12.42 (s, 1H, -COOH). HRMS (ESI+): *m*/*z*_calc_ for [M+Na]^+^ 298.13734, found 298.1379.

**Boc-Cit-PAB-OH:** Boc-Cit-OH (500 mg, 1.82 mmol, 1 eq.) and PAB-OH (448 mg, 3.64 mmol, 2 eq.) were dissolved in DCM:MeOH (2:1, *v*/*v*, 150 mL). EEDQ (897 mg, 3.64 mmol, 2 eq.) was added, and the reaction mixture was stirred at rt for 24 h under exclusion of light. Then, the solvent was removed under reduced pressure and the residue was triturated with diethyl ether (50 mL) for 5 h in an ultrasonic bath. After filtration, the solid was dissolved in DCM:THF (1:1, *v*/*v*) and loaded on a short silica column. Elution was performed with pure THF. The crude was purified via preparative RP-HPLC (+ 0.1% formic acid) and Boc-Cit-PAB-OH obtained as a pale-yellow oil. Trituration with DCM in an ultrasonic bath resulted in formation of a colorless solid (632 mg, 91%). ^1^H-NMR (500 MHz, MeOD) δ = 1.46 (s. 9H, Boc-H), 1.51–1.95 (m, 4H, -CH_2_-), 3.12 (m, 1H, -CH_2_-NH-), 3.21 (m, 1H, -CH_2_-NH-), 4.21 (m, 1H, Boc-NH-CH-), 4.57 (s, 2H, -Ph-CH_2_-OH), 7.32 (d, ^3^*J* = 8.11 Hz, 2H, Ph-H), 7.56 (d, ^3^*J* = 8.13 Hz, 2H, Ph-H). ^13^C-NMR (125 MHz, MeOD): 27.88 (-CH_2_-), 28.71 (-C(CH_3_)_3_), 30.93 (-CH_2_-), 40.34 (-CH_2_-NH-), 56.29 (Boc-NH-CH-), 64.84 (-Ph-CH_2_-OH), 80.65 (-C(CH_3_)_3_), 121.29 (Ph-C), 128.60 (Ph-C), 138.68 (Ph-C), 138.72 (Ph-C), 157.95 (-CO-), 162.34 (-CO-), 173.48 (-CO-). HRMS (ESI+): *m*/*z*_calc_ for [M+Na]^+^ 403.19519, found 403.1952.

**Boc-Cit-PAB-PNP:** Boc-Cit-PAB-OH (110 mg, 289 mmol, 1 eq.) and 4-nitrophenyl carbonate (264 mg, 868 mmol, 3 eq.) were dissolved in dry DMF (1.5 mL) and DIPEA (110 µL, 650 mmol, 2.25 eq.) was added. The reaction mixture was stirred under an argon atmosphere at rt for 5 h, followed by removal of the solvent under reduced pressure. Boc-Cit-Pab-PNP was obtained after column chromatography (5% MeOH in DCM) as a colorless solid (150 mg, 95%). R_f_: 0.2 (5% MeOH in DCM). ^1^H-NMR (500 MHz, CDCl_3_) δ = 1.38 (s, 9H, Boc-H), 1.44–1.76 (m, 4H, -CH_2_-), 2.92 (m, 1H, -CH_2_-NH-), 3.01 (m, 1H, -CH_2_-NH-), 4.09 (q, ^3^*J* = 7.62 Hz, 1H, Boc-NH-CH-), 5.25 (s, 2H, -Ph-CH_2_-O-), 5.41 (s, 2H, -NH_2_), 5.97 (s, 1H, -CH_2_-NH-), 7.04 (d, ^3^*J* = 7.84 Hz, 1H, Boc-NH-), 7.41 (d, ^3^*J* = 8.26 Hz, 2H, Ph-H), 7.57 (d, ^3^*J* = 8.96 Hz, 2H, Ph(NO_2_)-H), 7.65 (d, ^3^*J* = 8.20 Hz, 2H, Ph-H), 8.31 (d, ^3^*J* = 8.98 Hz, 2H, Ph(NO_2_)-H), 10.07 (s, 1H, -CO-NH-Ph-). HRMS (ESI+): *m*/*z*_calc_ for [M+H]^+^ 546.21946, found 546.2193.

**Boc-Cit-PAB-MMAE:** MMAE (15 mg, 20.9 mmol, 1 eq.) and Boc-Cit-PAB-PNP (13.7 mg, 25.8 mmol, 1.2 eq.) were dissolved in dry DMF (0.5 mL) under argon atmosphere. HOAt (8.5 mg, 62.4 mmol, 3 eq.) and DIPEA (10.65 µL, 31.4 mmol, 1.5 eq.) were added sequentially. The reaction mixture was stirred at rt overnight before it was directly purified via RP-HPLC (+ 0.1% formic acid). Boc-Cit-PAB-MMAE (quant.) was obtained as a colorless solid. LC-MS (ESI+): *m*/*z*_calc_ for [M+H]^+^ 1124.69, found 1124.72 (t_r_ = 9.0 min). HRMS (ESI+): *m*/*z*_calc_ for [M+H]^+^ 1124.69656, found 1124.6912.

**H-Cit-PAB-MMAE:** Boc-Cit-PAB-MMAE (16 mg) was dissolved in dry DCM (2 mL) and cooled to 0 °C. Dry TFA (2 mL) was added dropwise. The reaction mixture was stirred at 0 °C for 25 min and then further dry DCM (10 mL) was added. The solvent was removed under reduced pressure at 40 °C and the residue was dissolved in DMF (1.5 mL). H-Cit-PAB-MMAE (12 mg, 68%) was obtained after RP-HPC (+0.001% TFA) as a colorless solid. LC-MS (ESI+): *m*/*z*_calc_ for [M+H]^+^ 1024.64, found 1024.66 (t_r_ = 6.8 min). HRMS (ESI+): *m*/*z*_calc_ for [M+2H]^2+^ 512.82570, found 512.8239.


**H_3_N^+^-PEG_2_-Lys(mPEG_10_)-βAla-Val-OH:**


**Peptide synthesis:** Fmoc-Val-Wang resin with a loading of 0.71 mmol/g was used for the synthesis of the peptide. Coupling of Fmoc-βAla-OH, Fmoc-Lys(Mtt) and Boc-PEG_2_-CO_2_H was performed with fourfold excess of reagents (TBTU, DIPEA, amino acid) and automated and microwave assisted SPPS (CEM Liberty). Fmoc-cleavage was accomplished with 0.1 M HOBt in 20% piperidine in DMF (5 mL/g resin).

**Deprotection of Mtt protection group:** The resin was swollen in DMF and washed with DCM (10x). Then, the protected peptide was treated with 1% TFA in DCM (until the solution is decolorized), washed with DCM (5×), deprotonated with a solution of DIPEA in DCM (1:19, *v*/*v*) and washed again with DCM (5×) and DMF (5×).

**Coupling of mPEG_10_-CO_2_H:** The PEG-linker (125 mg, 1.66 eq.), DIPEA (40 µL, 1.66 eq.), HOAt (32 mg, 1.66 eq.) and HATU (81 mg, 1.5 eq.) were dissolved in dry DMF (1.5 mL) and incubated for 5 min. Then, the activated solution was added to the resin and the reaction mixture was shaken at rt overnight. After filtration, the resin was washed with DCM and DMF.

**Cleavage:** The cleavage was performed with TFA:H_2_O:TIPS (95:2.5:2.5, *v*/*v*/*v*) (8 mL) for 2 h at rt. After filtration, the solvent was removed under reduced pressure, the residue was dissolved in H_2_O (+ 0.1% TFA) and purified via RP-HPLC (+0.1% TFA). The pegylated peptide (56%) was obtained as a colorless oil.. LC-MS (ESI+): *m*/*z*_calc_ for [M+H]^+^ 974.58, found 974.69 (t_r_ = 4.9–5.5 min). HRMS (ESI+): *m*/*z*_calc_ for [M+Na]^+^ 974.57551, found 974.5750. ^1^H-NMR (500 MHz, ACN-*d_3_*) δ = 0.92 (d, ^3^*J* = 6.40 Hz, 3H, Val-C^γ^-H), 0.94 (d, ^3^*J* = 6.40 Hz, 3H, Val-C^γ^-H), 1.27 (m, 2H, Lys-H), 1.49 (m, 2H, Lys-H), 1.62 (m, 1H, Lys-H), 1.76 (m, 1H, Lys-H), 2.11 (m, 1H, Val-C^β^-H), 2.42 (m, 2H, Lys-H), 3.20 (m, 4H, βAla-H), 3.26–4.08 (m, 54H, PEG-H), 4.21 (d, ^3^*J* = 6.23 Hz, 1H, Val-C^α^-H), 4.33 (m, 1H, Lys-C^α^-H), 7.18 (d, ^3^*J* = 7.98 Hz, 1H, -NH-), 7.29 (m, 2H, -NH-), 7.44 (d, ^3^*J* = 8.02 Hz, 1H, -NH-).

**DBCO-PEG_2_-Lys(mPEG_10_)-βAla-Val-OH:** The previously synthesized peptide (60 mg, 61.5 mmol, 1 eq.) and DIPEA (28.1 µL, 184.6 mmol, 3 eq.) was dissolved in DMF (3 mL) and added to DBCO-NHS (28.3 mg, 65.7 mmol, 1.1 eq.). The reaction mixture was stirred at rt for 3 h before it was directly purified via RP-HPLC (+0.1% formic acid). The DBCO-modified peptide (44 mg, 55.5%) was obtained as a brown oil. LC-MS (ESI+): *m*/*z*_calc_ for [M+2H]^2+^ 645.35, found 645.35 (t_r_ = 7.9 min). HRMS (ESI+): *m*/*z*_calc_ for [M+2H]^2+^ 645.35436, found 645.35122.

**DBCO-PEG_2_-Lys(mPEG_10_)-βAla-Val-Cit-PAB-MMAE:** H-Cit-PAB-MMAE (12 mg, 10.5 µmol, 1 eq.), DBCO-PEG_2_-Lys(mPEG_10_)-βAla-Val-OH (17.67 mg, 13.7 µmol, 1.3 eq.), PyAOP (7.15 mg, 13.7 µmol, 1.3 eq.) and HOAt (4.3 mg, 31.63 µmol, 3 eq.) were dissolved in DMF (1 mL) and DIPEA (5.5 µL, 31.63 µmol, 3 eq.) was added. The solution was stirred at rt for 2 h and then half of it was purified directly by RP-HPLC (+0.01% NH_3_). The final linker (6.5 mg, 50.6%) was obtained as a colorless solid. LC-MS (ESI+): *m*/*z*_calc_ for [M+2H]^2+^ 1148.92, found 1148.68 (t_r_ = 4.9 min; isocratic method was used due to instability of DBCO during analysis). HRMS (ESI+): *m*/*z*_calc_ for [M+2H]^2+^ 1148.16750, found 1148.17078.

### 3.2. Biochemical/Biological Experiments

**hFGE expression:** The purification of human FGE was performed using the protocol published [29] with slight modifications. The recombinant human FGE was purified in two purification steps instead of three steps, which was published earlier. In brief, the recombinant human FGE was expressed as a secretory protein from High Five insect cells using a baculovirus expression system. The first step comprised dialysis of cell culture supernatant and then purification via a His-Trap column. In the second step, the purified fraction containing FGE was pooled and dialyzed against 20 mM Tris pH 8.0 and directly loaded on to a Mono Q column (GE lifesciences/Cytiva, Freiburg i. Br., Germany) skipping purification by size exclusion chromatography published earlier [29]. The bound FGE was eluted with a linear gradient to 1 M NaCl. The elution fractions were pooled and concentrated to a volume of about 250–500 µL using a Centricon concentrator (MWCO 10 kDa, Merck Chemicals, Darmstadt, Germany). The concentrated FGE was stored after flash freezing at –80 °C until further use.

**MtFGE expression:** Expression and purification of MtFGE from *E. coli* was performed exactly as described in our previous publications [35].

**DARPin monomer (DM) expression:** Expression and purification of monomeric DARPin E01 from *E. coli* was performed exactly as described in our previous publications [35].

**Design of DD1, DD2 and DFc:** Analysis of the crystal structure of dimerized ectodomains of the EGFR (PDB: 1IVO) revealed a distance of 72 Å between both Ala415 as essential residues of epitope mapping experiments for DARPin E01 [10,75]. The DARPin itself is a 45 Å long structure from N- to C-terminus. A (GGGGS)_4_ linker with a stretched length of approximately 64 Å was chosen as an interspersed linker for the single chain bivalent **DD1** and **DD2** construct similar to a previous design [45]. When forming a clamp, DARPins are in antiparallel orientation.

As a second design, a fusion to the N-terminus of the Fc domain of IgG1 was chosen in which the DARPins form a parallel oriented clamp due to Fc mediated homodimerization. A (GGSG)_2_ linker was introduced between the DARPin and the human IgG1 hinge region (DKTHC sequence) of the **DFc** resulting in a linker of 13 aa with a stretched length of 41 Å. Both genetically encoded linkers should enable bivalent binding to dimeric EGFR or otherwise clustered ectodomains.

**Design of scFvFc:** The scFv425 N-terminally fused to the human IgG1 Fc antibody fragment containing two FGE recognition motifs at the C-terminus is named Sc2 [39]. The Sc2 was modified by introducing the point mutation C487S inside the CTAGR-tag of the two C-terminal FGE recognition motifs to get the **scFvFc**.


**DARPin Dimer (DD2) Cloning, expression, and purification:**


**Cloning:** The commercially acquired plasmid pMA-RQ (see supporting information chapter 10) containing the DARPin Dimer (with N- and C-terminal CTPSR-tag and C-terminal His-tag) was digested with NdeI and HindIII and the fragment containing the gene of interest cloned in the likewise opened pET24b (Merck) using 1% agarose gel separation, gel extraction (QIAquick gel extraction kit, Qiagen, Hilden, Germany) and T4 ligase (NEB). Ligations were transformed into strain BL21(DE3). Clones were analyzed by expression and Western blot with anti-His-tag antibody and one positive clone (verified by sequencing) was stored at −80 °C as glycerol stock (43% (*v*/*v*) glycerol) and used for further experiments.

**Expression:** For heterologous protein production, a pre-culture was prepared overnight in 70 mL LB containing kanamycin. Next, the centrifuged cells were cultured in a 1.5-L main culture in an Erlenmeyer flask until an OD_600_ of 0.55 was reached. The culture was cooled to 18 °C in an ice bath and expression was started by adding IPTG (β-d-1-(isopropylthio)galactopyranoside) to a final concentration of 250 µM. Cells were incubated overnight, pelleted by centrifugation (10 min, 10.000 rpm, 12,857 rcf, 4 °C), resuspended in lysis buffer (50 mM phosphate, 300 mL NaCl, 25 mM imidazol, pH 8.0), supplemented with protease inhibitors (1 mM, cOmplete, Roche, Mannheim, Germany) and DNAse I (spatula tip, AppliChem, Darmstadt, Germany) and mechanically lysed using a French press (4 rounds 1000 psi, SLM Aminco, Urbana, IL, USA). Cell debris was removed via centrifugation (30 min, 10.000 rpm, 12,857 rcf, 4 °C) and filtration (pore size 0.22 µm, Filtropur S 0.2, Sarstedt).

**Purification:** The DARPin dimer **DD2** was purified by immobilized metal-ion affinity chromatography (IMAC) using a Ni^2+^-charged resin (HisTrap HP, 1 mL, GE lifesciences/Cytiva, Freiburg i. Br., Germany) and an automated system (ÄKTA explorer, GE GE lifesciences/Cytiva, Freiburg i. Br., Germany) (Appendix A). Proteins were eluted with a linear gradient of elution buffer: 50 mM phosphate, 300 mL NaCl, 250 mM Imidazol, pH 8. The desired protein was analyzed by SPS-PAGE (12.5%), pooled and rebuffered into PBS (10 mM Na_2_HPO_4_, 1.8 mM KH_2_PO_4_, 137 mM NaCl, 2.7 mM KCl, pH 7.4) using a Centricon (MWCO: 10 kDa, Sartorius, Göttingen, Germany). 38 mg purified protein based on absorption measurements (calculated 280 nm extinction coefficient: 44,920 M^−1^∙cm^−1^) was obtained per 1 L of cell culture. The enzyme was either used immediately or stored in PBS with 20 mM arginine at −80 °C until further use.

**Construction of plasmids coding for DD1, DFc or scFv425Fc:** The **DD1** coding sequence was amplified from pET24b_**DD2** via PCR by excluding the N-terminal CTPSR-tag using the primers 5′-TTTTTCATATGGATCTGGGTAAAAAACTGCTGGAAG-3′ and 5′-CACAGCTGCAGGATTTCTGCC-3′ and cloned in the pET24b backbone using NdeI and PstI resulting in the plasmid pZMB0787 coding for **DD1**. The coding sequence of **DFc** was cloned in three steps starting from pZMB0653 (pcDNA5/FRT_Sp-RAGE_DARPin-E01_IgG1-Fc_mCherry_Kp-AtsB). First, the sequence coding for a C-terminal part of the Fc domain was followed by a GGSGG-linker, the FGE recognition motif CTPSR, and a His_6_-tag was PCR amplified using the overhang primers 5′-AGCCCCGGGA ACCTCAGG-3′ and 5′-CCCTCCGCTC CCACCGATAT CAAAGGGCCC TCATGATGGT GGTGGTGATG TGCGGCCCGA GATGGGTGCA CAG-3′. The fragment was cloned in the backbone of pZMB0653 using XmaI and ApaI resulting in plasmid pZMB. In a second step a sequence coding for a (GGSG)_2_-linker was fused 5′ to hinge coding sequence upstream of the Fc sequence using two PCRs with the overhang primer pairs 5′-GCCACCTCGAGATTGTCGAGGTC-3′, 5′-CTGGAGGATCTCGGCCAGGTCGTGGGTCTTGTCACTG-ACCCACCGCCCGAGCCACC-3′ and 5′-ACCTGTCCCCCCTGTCCTGGGTGGCTCGG-GCGGTGGGTCAGGTGACAAGACCCAC-3′, 5′-TTCCCGGGGCTGGCCC-3′, which were subsequently combined with an overlap extension PCR. The final PCR product was cloned in pZMB0653 using XhoI and XmaI resulting in the plasmid pZMB. The C-terminus of pZMB0655 and the extended hinge region of pZMB0656 were combined via restriction cloning using XmaI and ApaI resulting in the plasmid pZMB0659 coding for **DFc**.

For **scFvFc**, a C-terminal fragment of the Fc part comprising the FGE dual-tag of the Sc2 from the plasmid pcDNA5/FRT_Sp-RAGE_scFv425_IgG1-Fc_dualFGE-tag (pZMB0520) [39] was PCR-amplified using the primers 5′-GGCGGGGAGCAAAGTTCTACCGCAGGTCGTGCTGCATTCATAACTGGGCAGGGTCTTTGCACACCATCACGTACCGGTTG-3′ and 5′-AGCCCCGGGAACCTCAGG-3′ (point mutation C487S) and cloned in the backbone of pZMB0520 using XmaI and AgeI resulting in the plasmid pZMB0720, which enables expression of the **scFvFc** with only one C-terminal aldehyde-tag.

**Protein expression and purification of DD1:** For **DD1** expression, BL21(DE3) cells were transformed with the plasmid pZMB0787 and cultivated for 16 h at 37 °C and 180 rpm. An aliquot was used to inoculate 0.5 L LB medium supplemented with 50 µg/mL kanamycin in a 2 L Erlenmeyer flask to a an OD_600_ of 0. The culture was cultivated to an OD_600_ of 0.6, when protein expression was induced with 0.1 mM IPTG, and after further cultivated at 18 °C for 16 h, cells were harvested by centrifugation (3220× *g*, 20 min) and stored at −20 °C. For protein purification, cells were thawed, resuspended in equilibration buffer (50 mM Na_2_HPO_4_, 300 mM NaCl, pH 7.4), and supplemented with 1 mM phenylmethylsulfonyl fluoride (PMSF). Cells were disrupted using a French press at a pressure of 1000 psi. Cell debris was pelleted by centrifugation at 15,000× *g* for 30 min at 4 °C. The supernatant was applied to a 1 mL Protino Ni-NTA (Macherey-Nagel, Düren, Germany) column for IMAC purification. After washing with 15 column volumes (CV) of equilibration buffer and 15 CV of equilibration buffer containing 24 mM imidazole the protein was eluted with 300 mM imidazole. Chromatography was monitored by measuring absorption at 280 nm and conductivity (Appendix A). Proteins were rebuffered in 50 mM bicine, 67 mM NaCl, 200 mM L-arginine, pH 9.0 using Amicon Ultra-4, MWCO 10 kDa (UFC8010, Merck Chemicals GmbH, Darmstadt, Germany) and stored at −80 °C. A batch of 1.5 L yielded in 65 mg **DD1** (Appendix A).

**Fc-fusion expression in mammalian cells and purification:** Transient expression of **DFc** or **scFvFc** in 60 mL HEK 293-F cells in suspension was performed in 250 mL shake flasks with orbital shaking at 185 rpm and 5 cm amplitude at 37 °C and 5 % COPrior to transfection, 293-F cells were washed with PBS. 1 mL cell suspension containing 3.0 × 10^6^ cells/mL was incubated with a DNA-PEI mixture of 2 μg DNA (pZMB0659 or pZMB0720) with 8 μg PEI MAX 40K (Polysciences, Hirschberg, Germany) in 250 μL HEK TF medium (Xell AG/Sartorius, Bielefeld, Germany) for 4 h before adding 1 mL HEK TF per 1 mL cell suspension. After five to seven days of cultivation, cells were harvested (2000× *g*, 5 min) and the cell culture supernatant was applied to a 1 mL HiTrap Protein A HP column (Cytiva) on an Äkta Start (1 mL/min, GE Life Sciences) chromatography system using as binding buffer 50 mM Tris, 150 mM NaCl, pH 8.0, as elution buffer: 50 mM Na_3_PO_4_, 50 mM citrate, 150 mM NaCl, pH 3.0, and as neutralization buffer 0.5 M Tris, 150 mM NaCl, pH 9. Up to 150 mL of sterile filtered cell culture supernatant was loaded at a time. The column was washed with a 15 CV binding buffer. Elution was performed with 10 CV elution buffer, collected in 0.5 mL fractions and neutralized with 180 µL of neutralization buffer. Chromatography was monitored by measuring 280 nm absorption and conductivity (Appendix A). Proteins were rebuffered in 50 mM bicine, 67 mM NaCl, 200 mM L-arginine, pH 9.0 using Amicon Ultra-4, MWCO 30 kDa (UFC8030, Millipore) and stored at −80 °C. Batches of 1 L and 700 mL yielded in 77 mg **DFc** and 52 mg **scFvFc**, respectively (Appendix A).

**FGly-conversion with hFGE:** The CTPSR-tagged protein (final concentration: 200 µM aldehyde-tag) was diluted with 5× bicine buffer (250 mM bicine, 333 mM NaCl, 1 M arginine, pH 9.3) (1/5 of V_total_) and water (V = V_total_ − (V_protein_ + V_bicin_ + V_DTT_ + V_FGE_)). Then, 100 mM DTT (final concentration: 3 mM for DARPin dimer and 4 mM for DARPinFc) and hFGE (18.5 mg/mL = 500 µM, final concentration: 30 µM, 15 mol%) were added sequentially, and the reaction mixture was shaken at 37 °C for 5 h. After removal of the precipitate, the protein was rebuffered into phosphate-buffer (50 mM phosphate, 50 mM NaCl, 1 mM EDTA, pH 6.7–7.0) using size exclusion desalting column (PD10; GE-Healthcare).

**FGly-conversion with MtFGE:** 100 mM DTT (final concentration 5 mM), non-reconstituted MtFGE (10 mg/mL, final concentration: 30 µM, 15 mol%) and 1 mM CuSO_4_ (final concentration: 30 µM, 15 mol%) were sequentially added to a solution of CTPSR-tagged protein (final concentration: 200 µM aldehyde-tag), 5x bicine buffer (250 mM bicine, 333 mM NaCl, 1 M arginine, pH 9.3) (1/5 of V_total_) and water (V = V_total_ − (V_protein_ + V_bicin_ + V_DTT_ + V_FGE_)). The reaction mixture was shaken at 22–25 °C for 16–20 h or at 37 °C for 5 h. Then, precipitated MtFGE was removed by short centrifugation and the protein was rebuffered via PD10 into phosphate buffer (50 mM phosphate, 50 mM NaCl, 1 mM EDTA, pH 6.7–7.0).

**Azide-functionalization by *tandem* Knoevenagel ligation:** A 250–500 mM stock solution of *tandem* Knoevenagel-azide in ACN:H_2_O (1:1, *v*/*v*) (final concentration: 10 mM) was added to the obtained FGly-proteins (DARPin monomer, DARPin dimer, DARPinFc, and scFv425Fc) and the mixtures were incubated at 30 °C for 20–24 h. Then, the protein was rebuffered into PBS using a desalting column (PD10, GE Healthcare).

**Click reaction with DBCO-Alexa-647, DBCO-PEG_3_-Val-Cit-PAB-MMAE and DBCO-PEG_2_-Lys(mPEG_10_)-βAla-Val-Cit-PAB-MMAE:** 15–120 µM of azide-functionalized proteins were treated with DBCO-Alexa-647, DBCO-PEG_3_-VC-PAB-MMAE, and DBCO-PEG_2_-Lys(mPEG_10_)-βAla-Val-Cit-PAB-MMAE (1–3 eq. based on azides) without dilution with buffer. In case of the MMAE conjugates, 1.5 equivalents per azide were used. The reaction mixtures were shaken at 37 °C for 16–24 h, and then the desired conjugate was purified directly by hydrophobic interaction chromatography.

**Hydrophobic interaction chromatography (HIC):** The conjugates were purified using a HiTrap Phenyl HP column (GE Healthcare) and an Äkta Ettan HPLC system (GE Life Sciences). The samples were diluted with binding buffer: 1 M (NH_4_)_2_SO_4_, 10 phosphate, pH 6.8 (for dye conjugates of **DD1**, **DFc** and **scFvFc**), pH 7.8 (for dye conjugate of **DM**) or pH 7.4 (for all drug conjugates). The mixtures were cleared by short centrifugation, and the supernatant was loaded on the equilibrated column at 0.5–1 mL/min. The column was washed with binding buffer (5–15 mL) and then proteins were eluted with a linear gradient of elution buffer: 10 mM phosphate, 30% isopropanol, pH 6.8 (for dye conjugates of **DD1**, **DFc** and **scFvFc**), pH 7.8 (for dye conjugate of **DM**) or pH 7.4 (for all other drug conjugates). 0.5 mL fractions were collected, and the absorption was measured at 280 nm. The product-containing fractions were combined, dialyzed (Spectra/Por 3 Dialysis Membran. MWCO: 3.5 kDa, SpectrumLabs/Repligen, Breda, Netherlands) overnight into PBS, and then concentrated with a Centricon (Sartorius, MWCO: 5 kDa). The labeling of proteins with Alexa-647 caused problems in concentration determination by Bradford assay or UV. Therefore, the concentration of the bound fluorophore was determined by absorption at 650 nm with ε = 270,000 cm^−1^ M^−1^. The concentrations of protein-MMAE conjugates were determined using Bradford assay with bovine serum albumin as standard.

**Analysis of conjugates by LC-ESI-ToF MS:** To determine the drug-to-antibody ratio (DAR), the conjugates were analyzed by LC-ESI-ToF MS using a reversed phase column (MAbPac RP 4 µm 2.1 × 50 mm, Thermo Scientific, Schwerte, Germany). Solvents were 0.1% formic acid in water and 0.1% formic acid in acetonitrile. Analyses by mass spectrometry were performed using a QToF mass spectrometer (Compact; Bruker Daltonics, Bremen, Germany). The spectra obtained were processed and annotated using DataAnalysis (Bruker Daltonics).

**Cell culture:** Isolation of the patient-derived glioma cell line BT12 has been previously described [52]. BT12 was maintained in serum-free DMEM/F12 medium supplemented with 1×B27 (both from Gibco, Thermo Fisher Scientific), 2 mM L-glutamine, 100 U/mL penicillin, 100 μg/mL streptomycin, 15 mM HEPES (all from Lonza, Cologne, Germany), 0.02 μg/mL human EGF, and 0.01 μg/mL human FGF-basic (both from PeproTech, Hamburg, Germany). A431 (ACC 91, DMSZ) and MCF7 (ACC 115, DSMZ) were cultivated in Roswell Park Memorial Institute medium RPMI 1640 supplemented with 10% fetal calf serum, 100 U/mL penicillin and 100 µg/mL streptomycin (all from Sigma-Aldrich). Human Dermal Fibroblasts adult (HDFa, C0135C, Thermo Fisher Scientific) and MDA-MB-231 (CRM-HTB-26, ATCC/LGC, Wesel, Germany) were cultivated in Dulbecco’s Modified Eagle Medium (DMEM) supplemented with 10% fetal calf serum 100 U/mL penicillin and 100 µg/mL streptomycin. The suspension cell line HEK FreeStyle 293-F (R79007, Thermo Fisher Scientific) was cultivated in HEK TF (Xell) medium supplemented with 8 mM L-glutamine.

**Live-cell imaging:** A431, MCF7 and MDA-MB-231 cells were cultivated in 8-well Nunc Lab-Tek or Lab-Tek II chambered coverglass with 2.5 × 10^4^ cells per well for 16 h at 37 °C and 5% CO_2_ atmosphere. For time dependent binding and internalization studies, cells were washed with RPMI or DMEM medium, incubated with 100 nM Alexa Fluor 647 dye conjugates for 10 min at 37 °C and 5% CO_2_, washed again and further incubated for 2 or 6 h at 37 °C and 5% CO. In case of lysosomal staining, cells were incubated with 50 nM LysoTracker Green DND-26 (Invitrogen, Thermo Fisher Scientific) for 10 min at 37 °C and 5% CO_2_ followed by washing with RPMI. To inhibit EGFR internalization, A431 cells were washed with RPMI, incubated with 100 nM Alexa Fluor 647 dye conjugates for 4 h at 4 °C and washed again. Cells were imaged at indicated time points starting from the protein incubation. MDA-MB-231 cells were imaged using a confocal fluorescence microscope (LSM 880, Zeiss, Oberkochen, Germany) with the objective Plan-Apochromat 63×/1.4 Oil DIC, MBS 488/561/633, laser 633 nm (5% power) with 643–758 nm emission detection (1.00 Airy unit) for the Alexa Fluor 647 fluorescence. A431 and MCF7 cells were imaged using a confocal fluorescence microscope (LSM 780, Zeiss) with the objective LCI Plan-Neofluar 63×/1.3 Imm Corr DIC, MBS 488/561/633, laser 633 nm (4% power) with 640–758 nm emission detection (1.35 Airy unit) for the Alexa Fluor 647 fluorescence and laser 488 nm (2% power) with 499–597 nm emission detection (1.73 Airy unit) by the 32-ch GaAsP detector for the DND-26 fluorescence. The line sequential scanning mode was used to minimize the effect of lysosomal movement for colocalization analysis. 12-bit images were processed with Fiji ImageJ 1.52p [76] and colocalization was analyzed by pseudo-coloring overlaying pixels.

**Penetration depth assay:** Cell suspension of BT12 spheres (passage 50) were incubated with 100 nM Alexa Fluor 647 dye conjugate to determine its penetration depth and 50 nM LysoTracker Red DND-99 (Invitrogen, Thermo Fisher Scientific) as counterstain for 30 min at 37 °C and 5% COCells were washed, transferred into an 8-well Nunc Lab-Tek II chambered coverglass (155409, Thermo Fisher Scientific) and further incubated in DMEM/F12 medium supplemented with 1× B27, 2 mM L-glutamine, 100 U/mL penicillin, 100 μg/mL streptomycin, 15 mM HEPES for 30 to 90 min at 37 °C and 5% CO. In total, five spheres (140–240 nm diameter) for each dye conjugate of **DM**, **DD1**, **DFc** and **scFvFc** were imaged using a confocal fluorescence microscope (LSM880, Zeiss) with the objective Plan-Apochromat 63×/1.4 Oil DIC, MBS 488/561/633, laser 633 nm (5% power) with 638–755 nm emission detection (1.00 Airy unit) for the Alexa Fluor 647 fluorescence and laser 561 nm (2% power) with 566–597 nm emission detection (1.17 Airy unit) for the DND-99 fluorescence in a line sequential scanning mode. The 12-bit images (224.92 µm × 224.92 µm, 2300 × 2300 pixels) were processed using Fiji ImageJ 1.52p [76] with the steps: 8-bit conversion, run plot profile of rectangle ROI (2300 × 300 pixels) through center of each sphere in x- and y-dimension of Alexa Fluor 647 channel. Penetration depth is defined as distance in µm of fluorescence signal (lower threshold 1.80) calculated from profile plot. The values of complete sphere penetration were bisected and considered only once for the analysis. Values from sphere margins outside the image frame were not considered. The two-sample t-test or Behrens-Fisher-test with a *p* < 0.05 level of significance was performed using the software R-4.1.1 to estimate statistical significance between categories.

**AlamarBlue assay:** The cytotoxicity of MMAE conjugates and free MMAE was assessed by the reduction of resazurin to fluorescent resorufin in metabolically active cells. 1 × 10^3^ adherent A431, MCF7 or HDFa cells were seeded in 96-well plates and incubated with **DD1**-MMAE, **DFc**-MMAE or free MMAE in a concentration range from 0.005 to 100 nM (stock diluted in cell culture medium) for 72 h at 37 °C and 5% CO_2_-atmosphere in a final volume of 100 µL per well. Next, cells were directly incubated with a final concentration of 0.02 g/L resazurin (add 10 µL from stock to cell culture medium) for 1 to 16 h at 37 °C. Metabolic activity is cell line dependent and A431 cells reduce resazurin faster than MCF7 cells. The fluorescence (ex. 560/9 nm, em. 590/20 nm) of resorufin was measured using a plate reader (Infinite M Plex, Tecan, Crailsheim, Germany). The OriginPro software (version 2021, OriginLab Corp., MA, USA) was used to fit dose response curves according to the function:(1)$$y=A_{1}+\frac{A_{2}-A_{1}}{1+10^{\left(\log x_{0}-x\right)p}}$$
*p* represents the hill slope. IC_50_ values are defined by the parameter log(𝑥_0_) and represent the mean of two biological replicates including three technical replicates for each cell line.

**Xenograft tumor models:** Animal studies were carried out according to the Animal Experiment Board in Finland (ELLA) for the care and use of animals under the license ESAVI/403/2019. Six-week-old female Rj:NMRI-Foxn1 nu/nu mice (*n* = 5 per treatment group, *n* = 1 per time period of dye conjugate, one inoculation per animal) were subcutaneously injected with 5 × 10^6^ A431 squamous carcinoma cells expressing high level of EGFR in 100 µL of PBS (Gibco/Thermo Fisher Scientific) into the lower right flank. Tumor size was followed two to three times per week using a vernier caliper measuring in three dimensions and tumor volume was calculated as the product of length × width × height × π/6. The mice were euthanized when tumors reached the ethical limits.

**Tumor targeting studies:** Nude mice bearing A431 tumors at a volume of 400–500 mm^3^ were intravenously injected into the tail vein with 150 µL of Alexa Fluor 647-labeled conjugate **DM**-AF647 (15 µM in PBS), **DD1**-AF647 (15 µM in PBS), **DFc**-AF647 (15 µM in PBS), or **scFvFc**-AF647 (10 µM in PBS) (*n* = 2 mice per group). The concentrations of the dye conjugates were determined according to the A_650 nm_ of the Alexa Fluor 647 chromophore from UV-Vis absorption measurements using a NanoDrop2000c. At each time point, whole body imaging was acquired using the Lago optical imaging system (ex. 605 nm, em. 690 nm, binning 4, exposure time 60 to 120 s), while mice were under isoflurane anesthesia (2.5% in 20% O_2_). After 6 or 96 h the mice were deeply anesthetized by intraperitoneal injection with 3.0 mg ketamine and 1.6 mg xylazine in 300 µL PBS to perfuse the animal via the left ventricle with 10 mL of ice-cold PBS followed by 10 mL 4% paraformaldehyde. Dissected tumors and organs were imaged ex vivo using the Lago optical imaging system (Spectral Instruments Imaging, Tucson, AZ, USA). Tumors and organs were fixed in 4% paraformaldehyde for 16 h at 4 °C and frozen in isopentane. For immunofluorescence analyses, the tumors were cut into 10 µm, the kidneys and livers into 9 µm sections using a CM1950 microcryotome (Leica Biosystems, Wetzlar, Germany), serried on microscope slides and stained with antibodies against EGFR and podocalyxin. Ex vivo images were quantified using Aura Imaging Software (Spectral Instruments Imaging, Tucson, AZ, USA) by determining the mean radiance in photons/s/cm^2^/sr of a 2 × 3 grid inside the tissue area. The mean radiance of each tissue was blanked with the corresponding value of each tissue of the PBS control group.

**Anti-tumor activity studies:** Nude mice bearing A431 tumors were intravenously injected into the tail vein twice weekly with 100 µL of drug-conjugate **DD1**-MMAE (6.5 mg/kg in PBS), **DFc**-MMAE (6.5 mg/kg in PBS), or PBS only with four injections in total (*n* = 5 mice per group). The treatment started at a mean tumor volume of 90 mm. The concentrations of the drug-conjugates were determined via Bradford assay. The tumors and selected organs were dissected, weighted and directly snap-frozen in isopentane. For immunohistochemical and immunofluorescence analyses, tissue sections were cut using a cryotome and serried on microscope slides. The tumors were cut into 10 µm sections and stained with hematoxylin and antibodies against Ki-67, EGFR and podocalyxin, and the kidneys and livers were cut into 9 µm sections and stained with antibodies against EGFR and podocalyxin. In addition, the tumor sections were analyzed for apoptotic cell death and necrosis with TUNEL (terminal deoxynucleotidyl transferase (TdT) dUTP nick-end labeling) assay. All statistical analyses were calculated using the software R. The two-sample t-test with a *p* < 0.05 level of significance was used to estimate statistical significance between categories.

**Immunohistochemistry:** For Ki-67 staining, snap-frozen sections (*n* = 5 sections from *n* = 3 tumors per group) were subjected to heat-induced epitope retrieval (HIER) in citrate buffer (0.01 M, pH 6.0). Sections were first incubated with 0.75% H_2_O_2_ for 10 min at 22 °C to block endogenous peroxidase activity. Sections were then incubated with a mouse monoclonal anti-human Ki-67 antibody (MIB-1,1:200; Dako/Agilent Technologies, Santa Clara, CA, USA) in Normal antibody diluent (Immunologic) for 30 min at 22 °C and detected using BrightVision Poly-HRP kit (Immunologic) and ImmPACT DAB Substrate Kit, Peroxidase (Vector Labs, Burlingame, CA, USA). Finally, sections were counterstained with hematoxylin solution, Harris modified (Sigma-Aldrich/Merck) and mounted with Aquatex (Merck, Darmstadt, Germany).

**Immunofluorescence:** Snap-frozen sections (*n* = 5 sections from *n* = 5 tumors per group) were fixed with 4% paraformaldehyde for 10 min at 22 °C, permeabilized in 0.3% Triton X-100 in PBS for 7 min at 22 °C and incubated in blocking solution A (10% fetal bovine serum, 0.03% Triton X-100 in PBS) for 1 h at 22 °C. Sections were incubated with the rabbit polyclonal anti-human EGFR (sc-03, 1:200; Santa Cruz Biotechnology, Santa Cruz, CA, USA) and monoclonal rat anti-mouse PODXL (MAB1556, 1:500; R&D Systems, Bio-Techne, Minneapolis, MN, USA) in blocking solution A for 16 h at 4 °C followed by secondary antibody incubation with Alexa Fluor 488 donkey anti-rabbit IgG (A32790, Invitrogen/Thermo Fisher Scientific, 1:500) and Alexa Fluor 594 donkey anti-rat IgG (A21209, Invitrogen/Thermo Fisher Scientific, 1:500) in blocking solution A for 1 h at 22 °C. The necrotic areas of the tissue sections (*n* = 1 section from *n* = 5 tumors) were quantified using Fiji ImageJ 1.52p [76]. Paraformaldehyde-fixed tumor, liver and kidney (*n* = 1 tissue/organ per group) sections (*n* = 5 sections per tissue/organ) from tumor targeting studies (6 h post-injection of dye-conjugate or PBS) were permeabilized in 0.6% Triton X-100 in PBS for 10 min at 22 °C and incubated in blocking solution B (10% fetal bovine serum, 0.06% Triton X-100 in PBS) for 1 h at 22 °C. Sections were incubated with the rabbit polyclonal anti-human EGFR (1:150) and monoclonal rat anti-mouse PODXL (1:375) in blocking solution B for 16 h at 4 °C followed by fluorescently labeled secondary antibody incubation (1:500) in blocking solution B for 1 h at 22 °C. Finally, sections were stained with DAPI (VECTASHIELD, Vector Laboratories) and mounted with Mowiol (Merck, Darmstadt, Germany).

**TUNEL assay:** For the TUNEL (terminal deoxynucleotidyl transferase dUTP nick end labeling) assay the In Situ Cell Death Detection Kit, TMR red (Roche) was used according to the manufacturer’s recommendations to label apoptotic and necrotic cells with DNA strand breaks. Briefly, snap frozen sections (*n* = 1 section from *n* = 3 tumors per group) were first fixed with 4% paraformaldehyde for 20 min at 22 °C, permeabilized with 0.1% Triton X-100 in 0.1% sodium citrate for 2 min on ice and then incubated with the TUNEL reaction mixture at 37 °C for 1 h. Finally, sections were stained with DAPI and mounted with Mowiol.

**Microscopy imaging:** Whole tissue sections were scanned using the 3DHISTECH Pannoramic 250 FLASH II digital slide scanner (3DHistech, Budapest, Hungary) at the Genome Biology Unit (University of Helsinki, Helsinki, Finnland and Biocenter Finland, a national network) using a Xenon flash lamp as light source and the filters DAPI-1160B (ex. 387/11 nm, MBS 409 nm, em. 447/60 nm, Semrock, Rochester, NY, USA), 38H GFP (ex. 470/40 nm, MBS 495 nm, em. 525/50 nm), 43HE Cy3 (ex. 550/25 nm, MBS 570 nm, em. 605/70 nm), 64HE mPlum (ex. 587/25 nm, MBS 605 nm, em. 647/70 nm) and Cy5-4040C (ex. 628/40 nm, MBS 660 nm, em. 692/40 nm, Semrock).

## 4. Conclusions

The modular conjugation system comprising the formylglycine-generating enzyme (FGE) and *tandem* Knoevenagel ligation in combination with strain-promoted azide-alkyne cycloaddition (SPAAC) was successfully applied for the preparation of homogenous protein-MMAE conjugates and thus represents a promising alternative to already known conjugation techniques. These conjugates were able to inhibit cell viability of epidermal growth factor receptor (EGFR)-overexpressing A431 cells with sub-nanomolar cytotoxicities and showed receptor-mediated endocytosis after specific binding of the receptor. Despite the successful in vitro results, no antitumor activities were observed in a subcutaneous squamous cell carcinoma model in vivo.

This study suggests that extremely high binding affinities of protein vehicles like DARPin E01 (K_D_ = 0.5 nM) are disadvantageous for the ubiquitously expressed tumor marker EGFR in an appropriate xenograft model [9]. Avidity is even further increased by bivalency. More important than the equilibrium dissociation constant K_D_ might be the dissociation rate constant k_off_. DARPin E01 is known for a low dissociation rate k_off_ = 0.2 × 10^−3^ s^−1^ DARPin E68 (K_D_ = 0.7 nM, k_off_ = 1.9 × 10^−3^ s^−1^) and E69 (K_D_ = 15.7 nM, k_off_ = 1.1 × 10^−3^ s^−1^) binding the EGFR ectodomain III and I, respectively, have five- to tenfold higher k_off_ rates [9]. This might be more promising for future in vivo tumor targeting due to better dissociation properties upon unspecific EGFR binding of normal tissues. The total of four administrations of 6.5 mg/kg **DD1**-MMAE or **DFc**-MMAE twice weekly are not causing any sequela. Therefore, low doses of unconjugated **DD1** or **DFc** could be suitable to saturate EGFR binding sites of normal tissues 6–24 h before injecting any anti-EGFR MMAE conjugate and push its anti-tumor efficacy. The predictive power of a mouse model for non-tumor binding in humans hinges on a given affinity of the binding molecule to the host receptor, which is often difficult to achieve for personalized approaches and not always reported. We benefitted from the analyses of the creators of DARPin E01, which is very valuable and a necessity for a target with ubiquitous expression.

Taken together, DARPins are a good choice for aldehyde coupling strategies due to their straightforward and high yield production as well as their high stability. As for antibodies, the affinity needs to be balanced if the target is also expressed on healthy cells. The conjugation strategy presented here is efficient and applicable to a wide range of targeting molecules.

## Data Availability

Data is contained within the article or Appendix A.

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
