# Peer review of "Bivalent EGFR-Targeting DARPin-MMAE Conjugates"

_ijms, 2022, doi:10.3390/ijms23052468_

Round 1
Reviewer 1 Report
Based on the anti-EGFR Designed Ankyrin Repeat Protein (DARPin), the authors have constructed the protein-drug conjugates DARPin dimer (DD1), DARPin-Fc (DFc), monomeric DARPin (DM), and antibody-derived scFv425-Fc (scFvFc) conjugated with monomethyl auristatin E (MMAE) using a PEGylated linker. They tested the cytotoxicity of DD1-MMAE, DFc-MMAE, and MMAE using A431, MCF7, and HDFa cells. DD1-MMAE and DFc-MMAE suppressed cell proliferation only in the A431 cells that had a high expression of EGFR; they did not suppress cell proliferation in EGFR-low MCF7 and HDFa cells. The authors also tested the anti-tumor efficacy of the conjugates using an A431-xenograft mouse model. Unfortunately, both DD1-MMAE and DFc-MMAE did not exhibit any anti-tumor efficacy. Although the manuscript puts forth interesting findings, there are some problems with it. The manuscript is currently not acceptable for publication in the high-impact International Journal of Molecular Sciences.
- Phosphorylation status is important for cell proliferation in EGFR-overexpressing cells. Did the authors determine the change in the phosphorylation status of EGFR upon in vitro or in vivo treatment with DD1-MMAE or DFc-MMAE?
- It was clear that DD1-MMAE and DFc-MMAE did not have any anti-tumor activity in the A431 cell-xenograft model (Figure 6A). Was a positive control, such as an MMAE treatment arm, used for this A431-xenograft model?
- Based on the figures shown, the difference in the anti-tumor activity between the in vitro experiments in A431 cells and in vivo experiments in the xenograft model using A431 cells was clear. Please describe the reasons why such differences might have occurred.
Author Response
Reviewer: 1
Based on the anti-EGFR Designed Ankyrin Repeat Protein (DARPin), the authors have constructed the protein-drug conjugates DARPin dimer (DD1), DARPin-Fc (DFc), monomeric DARPin (DM), and antibody-derived scFv425-Fc (scFvFc) conjugated with monomethyl auristatin E (MMAE) using a PEGylated linker. They tested the cytotoxicity of DD1-MMAE, DFc-MMAE, and MMAE using A431, MCF7, and HDFa cells. DD1-MMAE and DFc-MMAE suppressed cell proliferation only in the A431 cells that had a high expression of EGFR; they did not suppress cell proliferation in EGFR-low MCF7 and HDFa cells. The authors also tested the anti-tumor efficacy of the conjugates using an A431-xenograft mouse model. Unfortunately, both DD1-MMAE and DFc-MMAE did not exhibit any anti-tumor efficacy. Although the manuscript puts forth interesting findings, there are some problems with it. The manuscript is currently not acceptable for publication in the high-impact International Journal of Molecular Sciences.
A: Thank you for the general comment. DARPins are interesting protein vehicles for tumor therapy. However, for DARPins, especially the DARPin against EFGR, literature with in vivo studies is sparse. Therefore, our publication is an important contribution to the pharmacological properties of these proteins and helps other scientists to bridge the gap between in vivo and vitro. We feel that we can address the “problems” mentioned below.
- Phosphorylation status is important for cell proliferation in EGFR-overexpressing cells. Did the authors determine the change in the phosphorylation status of EGFR upon in vitro or in vivo treatment with DD1-MMAE or DFc-MMAE?
A: We did not examine the phosphorylation status of EGFR. The scope of this work is not on the effect of the targeting protein on the receptor, but on the synthesis of stable DARPin-MMAE conjugates and the selective transport of the FDA-approved drug MMAE to tumor cells, which is responsible for the antitumor activity demonstrated in in vitro cytotoxicity assays. However, phosphorylation status after DARPin E01 binding was previously studied in 2011 by Boersma et al (doi: 10.1074/jbc.M111.293266). According to these data, mono- and bivalent DARPin E01 constructs were able to inhibit phosphorylation of EGFR. We incorporated this important information in the introduction text.
- It was clear that DD1-MMAE and DFc-MMAE did not have any anti-tumor activity in the A431 cell-xenograft model (Figure 6A). Was a positive control, such as an MMAE treatment arm, used for this A431-xenograft model?
A: The A431-xenograft model has been an established model system for EGFR targeting in mice for decades. A positive control is only required when it is unclear whether the xenograft model used can be successfully targeted. We were able to demonstrate successful targeting by imaging cryo sections, thus a positive control was not necessary in our case. Furthermore, there is already a publication showing that MMAE has a mild inhibitory effect on A431 xenograft models (doi: 10.1002/1878-0261.12400) and the PK of MMAE in mice has been studied (10.3390/pharmaceutics11020098). Free MMAE would not be a good positive control, because it is not selective and there was no point in proving that our constructs is more efficacious than free MMAE. We added this information in the discussion of “2.3.3. In vivo efficacy of MMAE conjugates”.
- Based on the figures shown, the difference in the anti-tumor activity between the in vitro experiments in A431 cells and in vivo experiments in the xenograft model using A431 cells was clear. Please describe the reasons why such differences might have occurred.
A: Please find the enhanced detailed explanation in the Discussion of “2.3.3. In vivo efficacy of MMAE conjugates” and in the Conclusions.
Reviewer 2 Report
In the present study, the authors used the modular conjugation system comprising the formylglycine-generating enzyme (FGE) and tandem Knoevenagel ligation in combination with strain-promoted azide-alkyne cycloaddition (SPAAC) to successfully generate the homogenous protein-MMAE conjugates for the therapeutic application. They found that the conjugates were able to reduce the cell viability of EGFR-overexpressing A431 cells with sub-nanomolar cytotoxicities and showed receptor-mediated endocytosis after specific binding of the receptor. Despite the antitumor activities of the conjugates were not satisfied in a subcutaneous squamous cell carcinoma model in vivo, it is an interesting study. The authors provided valuable information to unravel the significance of the conjugates on therapeutic application. However, there are still several concerns described underneath.
Major points:
- In the present study, the protein-drug conjugates treatment dramatically inhibited the survival of A431 cells in vitro, but little reduce the tumor growth in vivo. Whether the authors determine the stability or turnover of the conjugates in vivo. The authors provide the information that DARPins are known to be cleared within minutes in Ref#45. However, Zahnd et al found that DARPins were very stable in PBS or serum within 4 weeks. Although the protein-drug conjugates are very different from DARPins.
- In figure 5, highly expression levels of anti-EGFR dye conjugates were accumulating in Kidney and liver post the anti-EGFR dye conjugates injection. Do Kidney and liver play critical roles in the clearance of drug conjugates or harbor highly expression of EGFR in these organs?
- Administration of the protein-drug conjugates show valuable potential to pass through the blood-brain-barrier into the brain. The protein-drug conjugates might be applied to deliver the specific drugs to specifically target metastatic tumor cells in brain in metastatic lung cancer or breast cancer patients.
Minor points:
- It is better to give a caption for every figure to summarize the results.
- The figure legends of figure 3 for human “derman” fibroblast (HDFa) should be corrected to dermal.
- There are several errors in the 3.2 Biochemical/biological experiments.
- The ref for hFGE expression should be Ref#29 not Ref#27. The authors should correct it.
- The information for DARPin monomer (DM) expression should be corrected. The description is repeated as one in MtFGE expression.
- The citations in the section of Design of DD1, DD2 and DFc should be followed with the style of references.
Author Response
Reviewer: 2
In the present study, the authors used the modular conjugation system comprising the formylglycine-generating enzyme (FGE) and tandem Knoevenagel ligation in combination with strain-promoted azide-alkyne cycloaddition (SPAAC) to successfully generate the homogenous protein-MMAE conjugates for the therapeutic application. They found that the conjugates were able to reduce the cell viability of EGFR-overexpressing A431 cells with sub-nanomolar cytotoxicities and showed receptor-mediated endocytosis after specific binding of the receptor. Despite the antitumor activities of the conjugates were not satisfied in a subcutaneous squamous cell carcinoma model in vivo, it is an interesting study. The authors provided valuable information to unravel the significance of the conjugates on therapeutic application. However, there are still several concerns described underneath.
Major points:
- In the present study, the protein-drug conjugates treatment dramatically inhibited the survival of A431 cells in vitro, but little reduce the tumor growth in vivo. Whether the authors determine the stability or turnover of the conjugates in vivo. The authors provide the information that DARPins are known to be cleared within minutes in Ref#45. However, Zahnd et al found that DARPins were very stable in PBS or serum within 4 weeks. Although the protein-drug conjugates are very different from DARPins.
A: Zahnd et al. determined the half-life of an unmodified DAPRin in mice to be about 3 min. There is no direct correlation between stability in buffer or serum and blood clearance. The clearance via the kidney is very size-dependent and tissue distribution depends on target abundance and affinity. The correlation between size and clearance has been studied e.g. by Anna Wu (doi: 10.1038/nbt1141). Thus, there is no contradiction between high stability and rapid excretion of DARPin E01. We added the properties of high serum stability and fast size-dependent blood clearance to the text in the discussion of “2.3.1 In vivo imaging of anti-EGFR dye conjugates” to further clarify this aspect.
- In figure 5, highly expression levels of anti-EGFR dye conjugates were accumulating in Kidney and liver post the anti-EGFR dye conjugates injection. Do Kidney and liver play critical roles in the clearance of drug conjugates or harbor highly expression of EGFR in these organs?
A: Yes, the liver and kidney play a significant role in the clearance of drug conjugates. In this process, the kidney serves as a kind of filter system that excretes low-molecular substances/proteins. Since our DARPin monomers or dimers are relatively small (< 50 kDa), the respective dye conjugates can be observed mainly in the kidney because of this clearance.
The liver is known to express EGFR (doi: 10.1093/nar/gky1056, doi: 10.1016/j.celrep.2017.04.048), so we expect some binding of DARPin E01 to the liver. Unfortunately, the k off rate of DARPin E01 is very low (see conclusion), even lower for bivalent constructs, leading to accumulation in the liver. We strongly hypothesize that this is EGFR binding in the liver, as our positive control - scFvFc - which only recognizes human and not murine EGFR, cannot be detected in the liver. Moreover, we can observe a stronger accumulation for the DARPinFc compared to the monomer, which can be explained by the higher avidity towards EGFR.
We clarified these observations and the corresponding conclusions of accumulation and binding acidity in the discussion of “2.3.2 Ex vivo imaging of anti-EGFR dye conjugates”. Further, we added the references for elevated EGFR expression in mouse liver.
- Administration of the protein-drug conjugates show valuable potential to pass through the blood-brain-barrier into the brain. The protein-drug conjugates might be applied to deliver the specific drugs to specifically target metastatic tumor cells in brain in metastatic lung cancer or breast cancer patients.
A: Unfortunately, in our ex vivo imaging we did not find relevant amounts of dye conjugate in the brain as semi-quantitative presented in the bar chart (Fig. 6H). So we strongly assume that the protein does not cross the brain-blood barrier. Furthermore, we see that our DARPins bind EGFR ubiquitously and thus did not reach the more easily accessible tumor in the skin tissue. Therefore, targeted treatment of brain tumors would not be possible with DARPin E01.
minor points:
- It is better to give a caption for every figure to summarize the results.
A: We have added the missing captions.
- The figure legends of figure 3 for human “derman” fibroblast (HDFa) should be corrected to dermal.
A: We have corrected this mistake.
- There are several errors in the 3.2 Biochemical/biological experiments.
A: We have corrected several mistakes.
- The ref for hFGE expression should be Ref#29 not Ref#27. The authors should correct it.
A: We have corrected this mistake.
- The information for DARPin monomer (DM) expression should be corrected. The description is repeated as one in MtFGE expression.
A: We have corrected this mistake.
- The citations in the section of Design of DD1, DD2 and DFc should be followed with the style of references.
A: We have corrected this mistake.
Reviewer 3 Report
please see attached report

Author Response
Reviewer: 3
First, the reviewer wishes to express their gratitude to the editors for giving them the opportunity to help the authors and to the authors for sharing their research.
I commend the authors for the thorough description of the methods used and the ample amount of supplementary information that expands and contextualises the findings of the main manuscript. I think this shows a significant effort towards documentation and reproducibility.
The key findings are that a labelling strategy presented in the paper can lead to the successful synthesis of DARPin-based drug and dye conjugates. However while these conjugates are successfully uptaken by cells in 2D culture and by spheroids and show enhanced toxicity in culture compared with the parental drug, this effect is not replicated in animal models. The authors suggest that this might be due to the binding affinity and kinetics, which cause the conjugate to bind non-specifically to EGFR expressed in off-target tissue and organs and suggest pre-loading with unlabelled DARPin or tweaking of binding kinetics as potential solutions to this problem. While this is a negative result, it may provide insight for other drug development projects.
Major points:
- Knoevenagel + SPAAC is proposed as an excellent alternative to existing conjugation methods, however of the five references proposed, the three that actually report any yields, report over 75% values. Based on yield alone, it is unclear, at least to me because I am not an expert in pharmacho-chemistry, what advantages the new conjugation method affords compared to the others. Maybe it would be worth expanding upon the matter?
A: As we wrote in the first sentence of Results and Discussion, the combination of Knoevenagel ligation and SPAAC is an improvement over the previously established aldehyde chemistry [38, 41, 42], as we were able to show in our previous publications [39, 40] (citations according to original submission). We obtained more homogeneous protein conjugates with this method, using much smaller amounts of the expensive drug-containing conjugation building block.
Compared to Anami et al [25], lower yields are obtained. However, their constructs are purified only by size exclusion chromatography, resulting in a heterogeneous protein mixture. Purification by hydrophobic interaction chromatography would have reduced their yield. We stated the advantages of our bioconjugation strategy at the end of “2.1.3. Protein-MMAE conjugates” again for clarification.
- The method used for assessing colocalisation is not really a quantitative method. I would suggest re-analysis of the data using one of the many available colocalisation plugins for ImageJ/FiJi and reporting either M1/M2 coefficients or at least a correlation coefficient, either in the main figure/text or in supplementary, in addition to the overlap map.
It is also unclear how many images have been acquired for each sample.
Also, in my experience it is super hard to see any binding of EGFR probes in MCF-7, the level of receptors is really only suitable for single-molecule experiments.
A: A table with corresponding coefficients analyzed with a common ImageJ plugin was added into the supplementary information. The live cell imaging experiment is not a quantitative, but a qualitative analysis of colocalization to simply state the observation of colocalization with lysosomes. MCF-7 cells were used as a negative control for EGFR-binding. It shows that each dye-conjugate retains the specific EGFR-targeting properties after chemical modification.
Minor points:
- Reference #21 has been incorrectly assigned to sortase, but it actually belongs to the transglutaminase group.
A: We corrected the mistake.
- In order to make the design process clearer to the non-specialist reader, it might be worth explaining briefly why increased local hydrophobicity of probes is detrimental and needs to be overcome.
A: We added a sentence.
- Only figure S12 refers to the differences between commercial and non-commercial DBCO linkers, while Fig S13-S14 refer to the conjugation of the various probes. It would be worth amending figure references to clarify this.
A: We corrected this.
- There is a typo in the first reference to the MCF7 cell line. The number is missing
A: We corrected this.
- The description of the labelling protocol (timings and temperature) is not entirely consistent with what reported in the legend of Figure 3 and in the methods. It might be a clarity issue rather than a consistency issue, but the description needs to be amended.
A: The legend of figure for and the description of the labelling protocol were clarified and are entirely consistent now.
- Figure S22 is never referred to in the text and it is never explained why you examined a longer timepoint compared to the leftmost column of fig 3A and what you conclude from the comparison. If the figure serves no purpose then perhaps it can be deleted.
Figures S24-25 should be a main text figure, really, as that is one of the key findings.
A: Figure S22 and the corresponding conclusion from it is now mentioned in the text. Furthermore, we joined Figure S24 and S25 together and moved it as new Figure 4 to the main text as suggested.
- Please state in the main text and the legend the number of mice per group and the dose of dye conjugate used. Presently it is only in the methods. A statement of where the injection was performed (I am guessing tail vein) might also be necessary for completeness.
There is a mistake in the caption of figure S32, AF657 instead of AF647
A: The number of mice (n= 2) per treatment group and the dose of each dye-conjugate was added to the legend and the main text. Intravenous injection into the tail vein is also stated now. The corresponding figure caption was corrected to AF647.
- The number of sections examined for each tumour is not stated and the reference to the supplementary figure for the Podocalyxin staining is wrong. It should be S39.
A: The missing information was added in the legend and the correct supplementary figure was listed.
- Please state the number of mice per group in the main text.
I am not sure what is the connection of the paragraph about TIF to the rest of the section. Perhaps clarify?
I am also not quite sure about the connection of last paragraph in the section, the one about EpCAM DARPins. I am not quite sure which point the authors are trying to make, that the binding/kinetic determinants of DARPin success in clinical models are currently unclear and should be investigated further?
A: The number of mice per treatment group was added in the main text. The error of the tumor volume due to the TIF by measuring with a vernier caliper was clarified in the main text.
There are only a few other in vivo studies of DARPin-drug conjugates with different targets, which are worth to mention. Our study points out the importance of the information, whether the protein vehicle is also binding to the corresponding murine antigen target. This contributes to the right judgement of anti-tumor efficacy data and their in vivo model. This relevant statement was added to the main text.
####den letzen Satz sollten wir ändern. Die Bindung an ubiquitär vorhandene Rezeptoren ist ja auch bei Antikörpern ein Problem. Letztendlich können wir nur vermuten, dass die Affinität an den murinen EGFR ausschlaggebend ist.
z.B. In in vivo studies on tumor targeting of anti-EpCAM DARPin showed promising results, yet the extent of cross reaction with the murine background could contribute to these findings.
Zuletzt kommt ein konstruktiver Ausblick als Schlusssatz beim Leser besser an (außer man möchte das Forschungsgebiet beerdigen). Z.B. Once the conjugation hurdle has been taken, finding the right binding molecule preferably targeting only a mutated, tumor specific receptor or providing an optimal balance of affinity will determine the success of ADCs.
- Could you please briefly describe the methods for hFGE, MtFGE and DM expression?
A: The description of hFGE expression was added. MtFGE and DM expression was exactly the same way performed as in our previous publications as cited in the method section.
- Three references in the Design of DD1, DD2 and DFc paragraph are incorrectly formatted
A: The reference style was corrected.
- In DD2 Purification, can you specify whether you used a gravity filtration or tangential flow filtration device?
A: A Centricon (Sartorius, MWCO: 10 kDa) was used. This specification was added to the text.
- In HIC paragraph, can you specify the MWCO of the Centricon filter?
A: The specification MWCO: 5 kDa was added.
- In Cell Culture section, the origin and accession number of the MDA-MB-231 cell line is missing. Additionally, in this section you state that you cultivated A431, MDA-MB-231 and MCF7 cells in RPMI 1640, however in the next section (Live-cell imaging) you also mention DMEM in relation to these cells. Please clarify which medium was used.
A: Origin and accession number of MDA-MB-231 was added. The statement about the cultivation medium was corrected to DMEM.
- Please add citation for ImageJ/Fiji and any plugins you have used which have associated publications. This is important for their funding.
A: The corresponding reference for Fiji ImageJ was added.
- In Anti-tumour activity studies, please state the number of mice per treatment group.
A: The number of mice per treatment group was added.
- In Immunohistochemistry Immunofluorescence and TUNEL assay, please state the number of sections assessed for each condition.
A: The numbers of sections were added for each condition.
- In Microscopy Imaging, please state the filters and illumination sources used to perform the imaging.
A: The filters and light source were added to the microscopy imaging section.
19.References #1, 2, 5 and 50 are very old and have been updated/superseded. Please use newer references.
A: References 1,5,50 have been updated. We kept the former reference [2], as it contains the EGFR crystal structure.
Round 2
Reviewer 2 Report
The revised manuscript is currently acceptable for publication in the International Journal of Molecular Sciences.